# Proximate Composition, Cyanide Content, and Carotenoid Retention after Boiling of Provitamin A-Rich Cassava Grown in Ghana

**DOI:** 10.3390/foods9121800

**Published:** 2020-12-04

**Authors:** Bright Boakye Peprah, Elizabeth Y. Parkes, Obed A. Harrison, Angeline van Biljon, Matilda Steiner-Asiedu, Maryke T. Labuschagne

**Affiliations:** 1Department of Plant Sciences, University of the Free State, Bloemfontein 9300, South Africa; brightpeprah@yahoo.com (B.B.P.); avbiljon@ufs.ac.za (A.v.B.); 2CSIR-Crops Research Institute, Kumasi 03220, Ghana; 3International Institute of Tropical Agriculture, Ibadan 200001, Nigeria; e.parkes@cgiar.org; 4Department of Nutrition and Food Science, University of Ghana, Accra 23321, Ghana; obedharrison@gmail.com (O.A.H.); tillysteiner@gmail.com (M.S.-A.)

**Keywords:** carotenoid retention, cassava biofortification, cyanide, protein content

## Abstract

Biofortified yellow-fleshed cassava is important in countries with high cassava consumption, to improve the vitamin A status of their populations. Yellow- and white-fleshed cassava were evaluated over three locations for proximate composition and cyanide content as well as retention of carotenoids after boiling. There was significant variation in the crude fiber, fat, protein and ash content of the genotypes. All but one of the yellow-fleshed cassava genotypes recorded higher protein values than the white-fleshed local genotypes across locations. The cyanide content of the genotypes varied between locations but was within the range of sweet cassava genotypes, but above the maximum acceptable recommended limit. Micronutrient retention is important in biofortified crops because a loss of micronutrients during processing and cooking reduces the nutritional value of biofortified foods. Total carotenoid content (TCC) ranged from 1.18–18.81 μg.g^−1^ and 1.01–13.36 μg.g^−1^ (fresh weight basis) for fresh and boiled cassava, respectively. All the yellow-fleshed cassava genotypes recorded higher TCC values in both the fresh and boiled state than the white-fleshed genotypes used as checks.

## 1. Introduction

Cassava roots are a staple food that provides carbohydrates and energy for more than 2 billion people in the world, while representing the main source of carbohydrate and energy for the approximately 700 million people living in the tropical and sub-tropical areas [1].

Vitamin A deficiency (VAD) is a widespread nutritional disorder in low-income countries, and is still a public health concern globally. VAD is the leading cause of preventable blindness in children. It leads to an increased risk of disease and death from diseases such as malaria, diarrhea and measles. [2].

Yellow-fleshed cassava genotypes rich in provitamin A (pVA), are part of the outputs of an international biofortification effort by HarvestPlus, the International Institute of Tropical Agriculture (IITA), the International Center for Tropical Agriculture (CIAT) and other national agricultural research institutions, to reduce vitamin A and other micronutrient deficiencies through the development of staple food crops with enhanced micronutrient content. Provitamin A biofortified cassava is genetically improved for increased provitamin A content. In the case of this material, it was done through conventional breeding. Replacing the white-fleshed cassava varieties grown by most farmers with new high pVA (yellow) cassava varieties to address micronutrient and health needs of people, could benefit an estimated 20 million children under 6 years of age, who are currently at risk from diseases associated with VAD. Biofortified staple crops with higher micronutrient density, including yellow-fleshed cassava varieties biofortified with pVA carotenoids, have been developed to improve food and nutrition security reducing micronutrient deficiencies across the world [3]. The United Nations have set 17 goals for Sustainable Development of which Goal 2 is the eradication of all forms of hunger, including hidden hunger, which refers to micronutrient deficiency [4]. Biofortified crops contribute directly to this goal.

In 2013, 15 yellow-fleshed cassava genotypes with total carotenoid content (TCC) levels between 4–18 μg g^−1^ were obtained from IITA-Nigeria. With the objective of releasing these genotypes, the Crops Research Institute (CRI) has been testing their agronomic performance across the various agro-ecologies in Ghana. Good cooking quality is an important parameter in selecting cassava for human consumption. Other factors important for selection are hydrocyanic acid (HCN) content, starch, fiber, cooking time, flavor, consistency and cooked pulp texture [5].

The roots consist mostly of starchy flesh (80% to 90% by weight) with 60.3% to 87.1% consisting of water [5,6]. Moisture content is very important in the shelf life of cassava flour, since levels higher than 12% allows for microbial growth, which significantly reduces its shelf life [5]. In cassava flour, the moisture is much lower than in roots and was reported to vary from 9.2% to 16.5% [7,8,9,10]. Cassava contains very low levels of protein of about 1–3% on a dry mass basis [8] and between 0.4 and 1.5 g 100 g^−1^ fresh weight [11]. Cassava therefore has much less protein than cereals such as maize and sorghum, that have about 10 g protein per 100 g fresh weight [12]. Cassava plants are very valuable, as they produce more weight of carbohydrate per unit area than other staple food crops under comparable agro-climatic conditions. Unfortunately, the low protein content and high starch content is the reason for the low nutritional value of the roots. About 50% of the crude protein in the roots consists of whole protein and the other 50% of free amino acids [13].

The aim of the HarvestPlus program is the improvement of micronutrient content of crops to such an extent that it will impact on human nutritional and health status in a way that can be measured. Equally important is to ensure that the agronomic characteristics of the crop, such as yield and disease resistance, is not negatively affected. The process of developing biofortified crops include factors such as nutrient retention after harvesting, how much of the crop is consumed, and whether the biofortified crop is acceptable to the consumer. The bioconversion from pVA to retinol in the case of pVA rich foods (called bioavailability), is also an important factor. The mechanisms must also be in place for large-scale dissemination of the biofortified crop, which may differ in specific target countries [14]. Carotenoids are very sensitive to light, heat and physical handling, which leads to losses during the processing of yellow-fleshed cassava roots into commonly consumed products [15]. Total carotenoid retention is therefore largely dependent on specific genotypes and processing methods used to prepare products [16].

The pVA content target level for cassava, set to reach 50% of the estimated average requirement for children and pregnant women in the DRC and Nigeria, assumes that up to 50% of pVA content in peeled roots is lost during processing, storage, and cooking [17,18]. Carotenoid retention higher than 50% in boiled cassava has been reported in different studies [19,20]. A study in Kenya demonstrated that feeding 2–4 years old children with boiled yellow-fleshed cassava improved their vitamin A status [21]. Cassava in Ghana is mainly traded as either dry pieces of fermented cassava roots (*konkonte)*, that are milled into cassava flour to prepare *banku,* or as fermented cassava paste (*bankye mole)*, used to prepare *koko*. Cassava is also boiled and pounded with plantain to prepare *fufu*. Generally, *fufu* in Ghana is prepared by cooking peeled cassava in boiling water, whereas *chikwangue* is prepared by precooking and steaming fermented cassava paste [22,23]. In Nigeria, a study found that apparent carotenoid retention in *fufu* prepared with fermented cassava flour was 17–32%, but no information on true retention was presented [24]. The same study also found that apparent retention of carotenoids was 86–90% when *fufu* was prepared with a wet paste without a drying step. Another study in Nigeria reported true carotenoid retention between 12 and 36% when processing biofortified cassava roots into *fufu,* using fermented cassava paste without a drying step [15]. There is limited information on carotenoid retention in cassava in a country like Ghana.

Despite its nutritional and commercial benefits, cassava contains toxic substances that limit its utility, the most important being cyanogenic glucosides, which are responsible for the bitter taste of some cassava cultivars. [25]. Glucosides such as linamarin and the linamarase enzyme react when cassava cells are mechanically damaged during harvesting. They then release acetone cyanohydrin, and this then decomposes to release cyanide [26], either by hydroxyl nitrile lyase or spontaneously when the pH is higher than 5 [27]. Cassava cultivars are therefore classified into two major types: bitter and sweet [28] based on the cyanogenic content. “Sweet” cassava variety roots contain less than 50 µg g^−1^ HCN on a fresh weight basis, whereas those classified as “bitter” varieties may contain up to 400 µg g*^−^*^1^ HCN [29]. However, the level of cyanide in the cassava roots can be effectively reduced with different processing and fermentation methods [30].

Cyanide is stored in vacuoles of cassava cells, and is known to be more concentrated in leaves and the root cortex compared to root parenchyma [31]. Several neurological diseases, including ataxic neuropathy, cretinism, and xerophthalmia are seen in areas where cassava is the staple food, and this has been attributed to cyanide poisoning [32,33]. Cyanide can also cause thyroid disorders, goiter and stunting in children [34]. Cassava toxicity levels vary depending on altitude, geographic location, the period of harvesting, crop variety and seasonal conditions [35]. Several cases of cassava poisoning have been recorded in Nigeria, all resulting from improper fermentation and processing of cassava. Cyanide exposure of more than 50 µg g^−1^ caused symptoms such as headache, weakness, changes in taste and smell, irritation of the throat, vomiting, lacrimation, abdominal colic, pericardial pain and nervous instability [36].

Cyanide content of cassava is higher during drought periods due to water stress in the plant [37]. The response of cassava plants to water stress is a function of both the duration and severity of water deficit and the cultivar. Cyanogen is the most important toxic substance in cassava, which is formed because of enzymatic hydrolysis of linamarin and lostaustralin. Cyanogen increases during drought because of the “concentration effect” from reduced yields (which increases cyanide per mass), due to water stress. The naturally high cyanogenic glucoside content of bitter cassava varieties is further increased by water stress. Dry season (inter-seasonal dry spells) water stress, is similarly known to result in increased cyanogenic glucoside levels in cassava [37]. During the dry season, cassava cyanogen levels can increase by 9–10 times their normal levels [38]. In Mozambique, more than 55% of fresh sweet roots became extremely toxic during drought periods, a trend which was also observed in other countries in Africa [31]. Cassava must, therefore, be processed to make it safe for consumption. Numerous processing techniques are used in cassava consuming countries. These techniques often improve palatability, extend shelf life, but also decrease the cyanogenic potential of cassava [39].

The aim of this study was to determine the TCC, proximate values and HCN in yellow-fleshed cassava genotypes and to measure the retention of carotenoids during the processing of biofortified cassava into boiled cassava. This will help breeders to identify genotypes with the best nutritional quality across the tested locations for planting and promotion.

## 2. Materials and Methods

### 2.1. Varieties, Field Trials and Sample Preparation

Ten cassava genotypes were evaluated, of which eight were selected from sets of yellow-fleshed cassava clones previously acquired from IITA, one commercial white-fleshed variety and one white-fleshed landrace obtained from farmer fields in Ghana (Table 1). Trials were conducted from May 2015–May 2016 at three locations situated in different agroecological zones. Fumesua (forest), Cape Coast (rain forest) and Ohawu (coastal savannah). Trials were laid out in a randomized complete block design with three replications, each consisting of four rows of five plants, giving a plot size of 20 plants per replication, therefore 60 plants per entry in total per trial, and 600 plots per trial for the 10 entries. This was repeated for three locations, giving a total of 1800 plots. Planting was done at a spacing of 1 × 1 m. Replications were separated by 2 m alleys. Weeding was done when necessary and experiments were entirely rain-fed. The soils for the trial sites at Fumesua are Asuasi series, a ferric acrisol with sandy loam topsoil over sandy clay. At Cape Coast, it is Benya series, Acrisol with deep yellowish-red to yellowish-brown, well to moderately drained alluvial clays and Ohawu has Toje-Alajo series, loamy topsoil over sandy loam soil. Annual rainfall for the sites during the trial period was Fumesua (1205 mm), Cape Coast (1295 mm) and Ohawu (1024 mm). The experimental areas have a bimodal rainfall pattern with the major rainy season in March to July and the minor rainy season from mid-August to November. There is a long dry harmattan season from December to March.

Plants from the two middle rows of each plot were harvested and bulked, and, from this, five roots were sampled. The samples were immediately transported to the laboratory, before they were washed and peeled. Samples from the apical, middle and distal portions of the roots were cut into small cubes of 10 mm^3^ and then bulked. Samples from the bulk were then heat-sealed in laminated bags of 1 kg each, and stored in a cool place until used. A sample of 30 g of these 10 mm^3^ pieces of fresh cassava roots were weighed and cooked, totally submerged, in 500 mL of boiling water for 20 min. A total of 60 samples (30 fresh samples and 30 boiled samples obtained from the ten genotypes with three replications) from each location were analyzed for various characteristics. For TCC and HCN only one replication per entry per location was analyzed, due to cost considerations.

### 2.2. Sample Analysis

Proximate analysis*:* Moisture and dry matter content, crude protein (Kjeldahl method), crude fat (Soxhlet method), crude fiber and ash content, were all determined using the AOAC methods [40]. All measurements were done in duplicate.

Determination of carbohydrate content: Total percentage carbohydrate was determined by adding the total values of crude protein, crude fat, crude fiber, moisture and ash constituents of the sample and subtracting it from 100% [41].

Determination of the cyanogenic potential of the fresh roots: An alkaline titration method [42] was used to measure HCN. The HCN concentration in each sample was calculated by multiplying the average titre value by the standard concentration of HCN released, which is 21.6 [43].

Determination of total carotenoid content using the spectrophotometer: One gram of each fresh sample of cassava cubes was weighed and ground using a mortar and pestle. Pyrogallol was added to facilitate the grinding and to prevent oxidation. Methanol (25 mL) was then added and the mixture was transferred by vacuum filtration into a conical flask corked with filter paper. Acetone (25 mL) was added to the residue to ensure maximum extraction of carotenoids. The volume of the filtrate was recorded as the volume after the first extract. The filtrate was then poured into a separating funnel (where the tap was closed) fixed to a retort stand. Petroleum ether (20 mL) was placed into the separating funnel before the extract was added. Distilled water was finally added. A separation of yellowish-colored organic solvent and the colorless inorganic solvent was observed. The tap of the separating funnel was opened for the inorganic solvent to run out into a beaker placed under the funnel, and finally discarded. Distilled water was again added to the sample for washing. The inorganic solvent was collected and discarded. Washing was repeated several times until the carotenoid solution was clear. All excess distilled water was discarded. A funnel was then placed in a conical flask under the separating funnel and the organic solvent containing the carotenoids was collected. The volume of the organic solvent was recorded as the volume of the second extract. A glass cuvette was filled with the organic solvent extract and the absorbance was read at 450 nm. The extraction was read in triplicate using a T80 UV/VIS spectrophotometer. TCC was then calculated using the following formula [44]:$$\mathrm{TCC}\;\left(µg\;g^{-1}\right)=\;\frac{A\;\times V\;\left(\mathrm{mL}\right)\;\times \;104}{A^{1\%}1\;\mathrm{cm}\times P\;\left(g\right)}$$
where A = is the absorbance, V = Total extract volume after second extraction P = Sample weight, A^1%^1 cm = 2592 (beta carotene extinction coefficient in petroleum ether).

The TCC of the boiled root samples was also determined using the spectrophotometry method. One gram of each boiled sample was ground using a mortar and pestle. The same procedure was then followed as for the raw samples.

### 2.3. Data Analysis

Data were subjected to analysis of variance (ANOVA) using SPSS, version 21 (https://www.ibm.com/support/pages/spss-statistics-210-available-download). Results were presented as means ± standard deviations, except for HCN and TCC where only one replication per entry per trial was analyzed. Differences between means were considered significant at *p* < 0.05 using the Duncan multiple range test.

## 3. Results

### 3.1. Moisture Content

The moisture content (Table 2) ranged from 50.48% to 83.84% at Cape Coast, 54.8% to 80.07% at Fumesua and 56.31% to 90.43% at Ohawu. Genotype Husivi (local) recorded the overall highest moisture content per fresh weight at the Ohawu location. At Cape-Coast, genotype IBA085392 had the highest moisture content and Cape Vars the lowest. At Fumesua and Ohawu, genotypes IBA070539 and Husivi (local) had the highest moisture content, respectively.

### 3.2. Carbohydrate Content

Carbohydrate content of samples from Cape-Coast, Fumesua and Ohawu ranged from 12.85% to 45.79%, 14.90% to 40.41% and 6.85% to 38.82% respectively (Table 2). The highest value was recorded for genotype Cape Vars (white-fleshed) across the three locations. Husivi (white-fleshed) and improved variety (Cape Vars) recorded higher carbohydrate content than most of the yellow-fleshed cassava genotypes. Ohawu had the highest mean moisture content and consequently the lowest carbohydrate content.

### 3.3. Protein and Fat Contents

Protein content ranged from 0.01 to 1.45% with genotype IBA070557 recording the highest value among the samples from Cape Coast, followed by genotype I090151, which recorded 1.32% and 1.26% at Fumesua and Ohawu respectively (Table 3). The protein content of the white-fleshed variety (Husivi) was generally lower than most of the yellow-fleshed cassava genotypes across all locations. Fat content ranged from 0.05% to 1.24% with genotype I070539 recording the highest value at Cape Coast and Fumesua, followed by genotype I070557 at Ohawu (1.14%). Generally, samples from Ohawu recorded the highest protein content, followed by those from Fumesua and Cape Coast. For fat content, genotypes from Cape Coast had the highest, followed by Fumesua and Ohawa.

### 3.4. Crude Fibre and Ash Content

Crude fiber is the part of food made up of cellulose and lignin. Genotype IBA083724 (2.62 %) had the highest fiber followed by IBA085392 and I083774 (2.57%), all at Fumesua (Table 4). Genotypes from Fumesua recorded the highest crude fiber (more than double that of Cape Coast), followed by those from Ohawu and Cape Coast.

Ash content is indicative of inorganic constituents (such as minerals) and for cassava, it generally ranges from 1% to 2%. Genotype IBA083724 (Ohawu) had the lowest ash content of 0.02% and is therefore likely to contribute less minerals in the diet when consumed. Almost all the yellow-fleshed cassava genotypes had higher ash content relative to the white-fleshed variety (local). Generally, genotypes from Cape Coast had the highest ash content, followed by those from Fumesua and Ohawa, but the mean values were similar.

### 3.5. Hydrogen Cyanide Content

The highest and lowest HCN content of the fresh cassava samples from Cape Coast was 47.76 µg g*^−^*^1^ and 23.88 µg g^−1^ in Cape Vars and IBA083774, respectively (Table 5). The HCN of samples differed significantly (*p* > 0.05). The highest HCN content was found in genotype IBA085392 (47.8 µg g*^−^*^1^) and the lowest in IBA090090 (23.9 µg g^−1^) in Ohawu. For Fumesua, the highest HCN content (43.1 µg g^−1^) was recorded in genotypes IBA070557 and IBA070593 with IBA085392 having the lowest value (9.9 µg g^−1^).

Across the three sites, genotypes Cape Vars from Cape Coast, IBA070593 and IBA070557 both from Fumesua, and IBA085392 from Ohawu had the highest HCN levels of 47.8 µg g*^−^*^1^, 43.1 mg kg*^−^*^1^ and 47.8 mg kg*^−^*^1^ respectively (Table 5). Genotypes IBA083774, IBA085392 and IBA090090 from Cape Coast, Fumesua and Ohawu, respectively, had the lowest HCN values of 23.9 mg kg*^−^*^1^, 9.9 mg kg*^−^*^1^ and 23.9 mg kg*^−^*^1^. Ohawu recorded the highest mean value for HCN (36.40 mg kg*^−^*^1^), followed by Cape Coast (30.69 mg kg*^−^*^1^) with the lowest from Fumesua (29.73 mg kg*^−^*^1^). There were no statistically significant differences in HCN levels across the three locations. Genotype Cape Vars recorded the highest mean value (39.63 µg g*^−^*^1^) for HCN, followed by genotypes IBA070557 (36.00 µg g*^−^*^1^) and IBA070593 (36.00 µg g*^−^*^1^) across all three locations. Genotype IBA083724 recorded the lowest HCN mean (25.77 µg g*^−^*^1^).

### 3.6. Total Carotenoid Content

The color of the cut cross-section of fresh roots show color ranges that depicts the level of TCC, this is visually assessed by color chat. Actual levels are determined in the laboratory. For this study, the concentration of TCC in the fresh roots ranged from 1.18 µg g^−1^ (Cape Vars) for samples from Cape Coast to 18.81 µg g*^−^*^1^ (I070539) for samples from Ohawu (Table 6). For the boiled analysis, TCC ranged from 1.01 µg g*^−^*^1^ (Cape Vars) for samples from Cape Coast to 13.86 µg g*^−^*^1^ (I083724) for samples from Fumesua. Boiling was found to decrease the total carotenoid content of the different genotypes across all three locations. For both boiled and fresh samples, there were differences in TCC content across the three locations. Fresh samples in Fumesu recorded the highest values, followed by Ohawu and then Cape Coast (10.71 ± 4.27 µg g*^−^*^1^, 10.61 ± 4.27 µg g*^−^*^1^, 5.87 ± 3.16 µg g*^−^*^1^; *p* = 0.02 respectively). The same trend was observed after boiling.

## 4. Discussion

The results of the proximate analysis of the different cassava genotypes samples from three locations revealed wide variation for all traits with ranges of 50.48–90.4% for moisture content, 6.85–45.79% for carbohydrate, 0.01–1.26% for protein, 0.07–1.24% for fat, 0.47–2.62% for fiber and 0.37–2.34% for ash. Significant differences (*p* < 0.001) were found amongst the genotypes for each of the proximate analysis parameters. In general, the observed ranges were below values reported previously [45,46]. The maximum limit for the crude fiber and fat content observed agreed with values reported previously [47]. However, the values observed for fat content was higher than the values reported previously [36].

The carbohydrate values obtained in this study were lower than values reported previously, which had a range of 62.0–72.4% [48], 87–89% [49] and 85–89% [45]. The results in this study generally indicated that yellow-fleshed cassava tends to have less carbohydrate than white-fleshed varieties.

Crude ash content is usually indicative of inorganic constituents (minerals such as K, Zn and Ca) and for cassava, and generally ranges from 1% to 2%. Ash contents represents the total mineral content in food after it has been burned at a very high temperature. The ash and protein contents were lower than values reported by other studies [45,50], but were similar to those reported by others [32,45] from six yellow and white cassava varieties cultivated in Umudike, Nigeria.

Cyanide concentrations vary in different cassava genotypes according to the altitude, geographical location and seasonal and production conditions [51]. Reports have shown that age, variety and environmental conditions influence the occurrence and concentration of cyanide in various parts of the cassava plant and at different stages of development [7], hence the genotypes need to be tested at different ages of maturity for further inferences. Cassava is classified as sweet if cyanide content is less than 50 µg g*^−^*^1^ or bitter if the total cyanide is more than the 50 µg g*^−^*^1^. In drought conditions, there is an increased total cyanide content due to water stress [29]. Thus, a variety is considered to be “sweet” under one set of conditions may be “bitter” in a different geographical location or climatic conditions [43]. Values from 15–400 µg g*^−^*^1^ fresh weight of total cyanide in cassava roots have been reported in different studies, and there were reports of even higher levels, depending on where the crop was grown [29,51]. However, the rates can be reduced in cassava with different processing and fermentation methods. The observed levels of cyanide obtained in the present study showed that all the genotypes sampled could be classified as sweet varieties. The values were lower than those reported previously [25] but it is not advisable to eat it raw, since the range is above the acceptable limit (10 µg g*^−^*^1^).

A loss of micronutrients during processing and cooking is undesirable, as it reduces the nutritional value of biofortified foods. It is therefore particularly important that biofortified crops should be able to retain sufficient levels of micronutrients after typical processing, storage and cooking practices for successful biofortification.

All eight tested yellow-fleshed genotypes had TCC higher than the farmer preferred variety (Husivi) and the improved check (Cape Vars) in both fresh and boiled states. TCC of freshly peeled cassava of the evaluated genotypes was 1.18 to 18.81 µg g*^−^*^1^ on fresh weight basis, whereas in boiled cassava it was lower, with a range of 1.01 to 13.36 µg g*^−^*^1^ across the three locations. Large genetic variation for TCC in 12 yellow cassava genotypes in Brazil was also reported [52]. Genetics was also reported to be a large contributor to variation in yellow and white cassava and their products [53]. In general, there was a decrease in TCC level during boiling. The average TCC loss was 27.26% for Cape Coast, 15.03% for Fumesua and 16.59% for Ohawu, but there was large variation between genotypes for TCC loss, for example, I090151 lost only 7.96% TCC across locations, while I070557 lost 31.23% (Table 6). This is in contrast with previous findings [54] that carotenoid retention was better when sweet potatoes were boiled for the shortest possible time compared to methods like drying, frying and roasting that caused reduced retention. Another study [13] also reported that boiling lead to higher TCC retention compared to other processing techniques in sweet potatoes. A study on yellow cassava in Colombia [55] showed a mean retention of 87% of carotenoids after 30 min of boiling, which was higher than the 80.4% average in the current study. Their study also showed that dry matter content after boiling influenced TCC, and should be considered when measuring TCC retention. Another study [56] showed a much lower TCC retention of 47.87–83.79% in three yellow cassava genotypes after 10 min of boiling. Both these studies indicated that initial TCC influenced retention after boiling, therefore genotypes that had high TCC before boiling also had the highest TCC after boiling. This was generally also the case in the current study, where genotypes that had high TCC before boiling, ranked high for TCC after boiling. Contrary to these studies, almost no influence of boiling for 30 min on β-carotene content in yellow cassava (on a dry weight basis) for both fufu and boiled cassava was reported in another study [57].

Different factors separately or combined, such as heat, light, oxygen and enzymes, can lead to major or minor losses of carotenoids in yellow cassava during processing into consumable products [18,58]. The losses observed in the study for boiled roots could also be due to carotenoid isomerization and oxidation, which is the breakdown of trans-carotenoid to their cis-isomers due to increased content with moisture and heat treatment during boiling [59]. Gari is also one of the most popular products of cassava processing in Ghana and sub-Saharan Africa, and it has been reported that extended roasting during its processing results in higher carotene content [19]. Gari may therefore, be a useful way of efficiently utilizing biofortified cassava in VA deficient population. Further studies on more varieties commonly used for cassava dough, fufu, konkonte and gari may be needed to ascertain how yellow-fleshed cassava varieties may respond for TCC during processing into Ghanaian food forms.

Previous findings [60,61,62] confirm TCC loss patterns in cassava products consumed in sub-Saharan Africa. The result further suggests that the current yellow-fleshed cassava genotypes being evaluated could provide more VA in diets and contribute to the reduction of health challenges associated with VAD, which is widespread in Ghana and sub-Saharan Africa. Following the agricultural transformation agenda in Ghana (Modernization of Agriculture in Ghana), which has resulted in the availability of improved varieties (including biofortified cassava), there is a great need to scale up micronutrients in staple foods produced in the country [63]. Even though the impact of consuming yellow flesh cassava products on VA serum concentration is not yet fully established in VAD populations in Ghana, the results give an indication that yellow-fleshed cassava varieties are better than white-fleshed ones in terms of carotenoid and protein contents, and have the potential of reducing VAD in Ghanaian populations, where it is still endemic.

## 5. Conclusions

Given the importance of cassava to the economy of Ghana, and its role as a major crop in alleviating hunger in Africa, genetic improvement of the crop to address food and nutritional needs of Ghana and the continent has been recognized. The crop has to be improved for productivity, proximate composition and safe cyanide content. The yellow-fleshed cassava varieties evaluated have better TCC levels than varieties grown by Ghanaian farmers. With increasing awareness on the toxicity of cyanide in cassava, and given that cassava is mainly consumed as processed food, yellow-fleshed cassava is safe for consumption as food to enhance vitamin A. Biofortification of cassava genotypes presents a viable and promising intervention for tackling VAD in disease-burdened populations of sub-Saharan Africa. The WHO is advocating food and nutrition security, and yellow-fleshed cassava could be a key driver for this purpose. Considering that cassava is consumed principally as processed foods, further studies are needed to ascertain the TCC losses during the processing of the various food forms the crop is consumed in. There is also the need to develop new farmer-preferred cassava varieties that combine high TCC with high DMC to increase their rate of adoption, since most of these varieties/landraces already possess some important traits like high DMC and stability, which are key drivers of cassava adoption. Further, breeding of cassava varieties with higher TCC is ongoing in Ghana and in many other countries. It is, therefore, expected that more varieties will be available in the near future with increased adoption rates and increased pVA intake.

## Figures and Tables

**Table 1 foods-09-01800-t001:** Yellow- and white-fleshed cassava genotypes used for the study.

Genotype	Code	Status	Source	Pulp Color
IBA090090	G1	Improved	IITA	Yellow
IBA090151	G2	Improved	IITA	Yellow
IBA070557	G3	Improved	IITA	Yellow
IBA085392	G4	Improved	IITA	Yellow
Husivi	G5	Landrace	Farmer	White
IBA083774	G6	Improved	IITA	Yellow
IBA070593	G7	Improved	IITA	Yellow
IBA070539	G8	Improved	IITA	Yellow
Cape Vars	G9	Released	CSIR-CRI	White
IBA083724	G10	Improved	IITA	Yellow

**Table 2 foods-09-01800-t002:** Percentage moisture and carbohydrate content of fresh cassava varieties from three different locations.

Moisture Content (%)				Carbohydrate Content (%)		
Variety	Cape-Coast	Fumesua	Ohawu	Cape Coast	Fumesua	Ohawu
I090090	70.40 ± 13.10 ^abcd^	76.60 ± 0.20 ^c^	70.30 ± 1.00 ^bc^	26.90 ± 13.10 ^abc^	20.10 ± 0.10 ^ab^	25.80 ± 0.80 ^de^
I090151	79.50 ± 10.30 ^cd^	66.90 ± 0.30 ^ab^	64.80 ± 2.00 ^b^	17.90 ± 9.80 ^ab^	29.00 ± 0.10 ^cd^	6.90 ± 1.90 ^ef^
I070557	66.90 ± 0.78 ^abcd^	62.20 ± 3.70 ^a^	66.90 ± 0.20 ^b^	30.70 ± 0.40 ^bcd^	34.20 ± 0.20 ^d^	27.30 ± 1.70 ^de^
I085392	83.80 ± 4.45 ^d^	64.70 ± 7.00 ^a^	69.70 ± 0.90 ^bc^	12.90 ± 4.40 ^a^	30.20 ± 9.50 ^cd^	27.20 ± 1.00 ^de^
I083724	58.20 ± 11.70 ^ab^	67.30 ± 0.30 ^ab^	76.60 ± 6.30 ^cd^	38.90 ± 11.80 ^cd^	28.50 ± 0.60 ^bcd^	19.90 ± 6.20 ^bcd^
I083774	66.20 ± 0.26 ^ab^	62.20 ± 3.70 ^a^	67.50 ± 5.00 ^b^	34.40 ± 0.10 ^bcd^	34.90 ± 1.30 ^d^	24.50 ± 10.90 ^cde^
I070593	66.80 ± 3.10 ^abcd^	73.30 ± 5.10 ^bc^	82.80 ± 3.50 ^de^	29.00 ± 5.70 ^abcd^	24.50 ± 5.10 ^bc^	15.00 ± 3.40 ^abc^
I070539	50.70 ± 2.00 ^a^	80.10 ± 0.50 ^c^	83.70 ± 4.30 ^cde^	45.60 ± 2.10 ^d^	14.90 ± 1.90 ^a^	13.00 ± 4.40 ^ab^
Cape Vars	50.40 ± 11.80 ^a^	54.80 ± 1.10 ^ab^	56.30 ± 5.70 ^ab^	45.80 ± 11.60 ^d^	40.40 ± 0.80 ^bcd^	38.80 ± 5.40 ^f^
Husivi	66.20 ± 2.60 ^abc^	63.10 ± 0.20 ^a^	90.40 ± 2.00 ^a^	30.70 ± 2.60 ^bcd^	34.30 ± 0.20 ^d^	33.00 ± 2.60 ^a^
Mean	65.91	67.12	72.90	31.28	29.10	23.14
*p*-value	<0.01	<0.01	<0.01	<0.01	<0.01	<0.01

Values are presented as means and ± standard deviations. Means along the same column with different superscripts are statistically different (*p* < 0.05).

**Table 3 foods-09-01800-t003:** Protein and fat percentage of cassava varieties from three different locations.

Protein Content (%)				Fat Content (%)		
Variety	Cape-Coast	Fumesua	Ohawu	Cape Coast	Fumesua	Ohawu
I090090	0.24 ± 0.01 ^d^	0.28 ± 0.01 ^d^	0.45 ± 0.01 ^e^	0.92 ± 0.10 ^cd^	0.07 ± 0.04 ^a^	0.74 ± 0.10 ^cd^
I090151	0.32 ± 0.01 ^f^	1.32 ± 0.01 ^a^	1.26 ± 0.01 ^c^	0.72 ± 0.10 ^ab^	0.87 ± 0.30 ^cd^	0.05 ± 0.00 ^a^
I070557	1.45 ± 0.01 ^a^	0.58 ± 0.01 ^f^	0.85 ± 0.01 ^g^	0.94 ± 0.20 ^cd^	1.16 ± 0.04 ^d^	1.14 ± 0.50 ^d^
I085392	0.01 ± 0.01 ^a^	0.37 ± 0.01 ^a^	1.12 ± 0.01 ^h^	0.96 ± 0.10 ^cd^	0.27 ± 0.30 ^ab^	0.12 ± 0.03 ^ab^
I083724	0.31 ± 0.01 ^f^	0.25 ± 0.01 ^c^	0.67 ± 0.01 ^f^	1.05 ± 0.04 ^de^	0.12 ± 0.10 ^a^	0.47 ± 0.24 ^bc^
I083774	0.19 ± 0.01 ^c^	0.17 ± 0.01 ^a^	0.84 ± 0.01 ^g^	0.96 ± 0.03 ^cd^	0.74 ± 0.10 ^bcd^	0.84 ± 0.10 ^cd^
I070593	0.08 ± 0.01 ^b^	0.18 ± 0.01 ^b^	0.27 ± 0.01 ^c^	0.67 ± 0.03 ^ab^	0.40 ± 0.40 ^abc^	0.07 ± 0.03 ^a^
I070539	0.27 ± 0.01 ^f^	0.26 ± 0.01 ^c^	0.37 ± 0.01 ^d^	1.24 ± 0.10 ^e^	1.22 ± 0.02 ^d^	0.30 ± 0.10 ^ab^
Cape Vars	0.02 ± 0.01 ^a^	0.10 ± 0.01 ^a^	0.23 ± 0.01 ^b^	0.54 ± 0.00 ^a^	0.45 ± 0.30 ^c^	0.27 ± 0.10 ^ab^
Husivi	0.03 ± 0.01 ^a^	0.18 ± 0.01 ^b^	0.13 ± 0.01 ^a^	0.82 ± 0.03 ^bc^	0.17 ± 0.20 ^a^	0.05 ± 0.00 ^a^
Mean	0.29	0.37	0.62	0.88	0.55	0.41
*p*-value	<0.01	<0.01	<0.01	<0.01	<0.01	<0.01

Values are presented as means and standard deviations. Means in the same column with different superscripts are statistically different (*p* < 0.05).

**Table 4 foods-09-01800-t004:** Crude fiber and ash percentage of fresh cassava samples across three different locations.

	Crude Fiber (%)			Ash Content (%)		
Variety	Cape-Coast	Fumesua	Ohawu	Cape Coast	Fumesua	Ohawu
I090090	0.75 ± 0.10 ^ab^	2.53 ± 0.20 ^c^	2.07 ± 0.03 ^d^	0.72 ± 0.10 ^ab^	0.42 ± 0.10 ^a^	0.62 ± 0.30 ^ab^
I090151	1.04 ± 0.20 ^abc^	2.48 ± 0.10 ^c^	1.33 ± 0.20 ^ab^	1.02 ± 0.30 ^ab^	0.60 ± 0.40 ^a^	0.65 ± 0.40 ^ab^
I070557	0.47 ± 0.30 ^a^	1.24 ± 0.10 ^ab^	1.72 ± 0.04 ^c^	1.02 ± 0.70 ^ab^	0.59 ± 0.08 ^a^	2.11 ± 2.00 ^ab^
I085392	0.94 ± 0.21 ^abc^	2.57 ± 0.20 ^c^	1.49 ± 0.01 ^bc^	1.39 ± 0.10 ^ab^	1.94 ± 2.00 ^a^	0.37 ± 0.04 ^ab^
I083724	1.04 ± 0.50 ^abc^	2.62 ± 0.04 ^c^	2.23 ± 0.10 ^de^	0.82 ± 0.30 ^ab^	1.17 ± 0.30 ^a^	0.02 ± 0.00 ^a^
I083774	1.02 ± 0.10 ^abc^	2.57 ± 0.20 ^c^	1.47 ± 0.04 ^bc^	1.22 ± 0.30 ^ab^	0.47 ± 0.04 ^a^	2.34 ± 2.30 ^b^
I070593	1.34 ± 0.60 ^bc^	1.16 ± 0.40 ^a^	1.03 ± 0.04 ^a^	2.09 ± 1.90 ^b^	0.42 ± 0.10 ^a^	0.77 ± 0.04 ^ab^
I070539	1.42 ± 0.03 ^c^	2.53 ± 0.20 ^c^	1.72 ± 0.04 ^c^	0.72 ± 0.10 ^ab^	2.07 ± 2.40 ^a^	0.90 ± 0.00 ^ab^
Cape Vars	1.02 ± 00 ^abc^	1.74 ± 0.30 ^b^	2.54 ± 0.40 ^a^	0.72 ± 0.04 ^ab^	0.69 ± 0.30 ^a^	0.79 ± 0.00 ^ab^
Husivi	1.04 ± 0.50 ^abc^	1.65 ± 0.30 ^b^	2.10 ± 0.01 ^d^	0.52 ± 0.40 ^ab^	0.57 ± 0.04 ^a^	0.45 ± 0.10 ^ab^
Mean	1.01	2.11	1.77	1.02	0.89	0.90
*p*-value	<0.01	<0.01	<0.01	<0.01	<0.01	<0.01

Values are presented as means and standard deviations. Means in the same column with different superscripts are statistically different (*p* < 0.05).

**Table 5 foods-09-01800-t005:** Comparison of hydrogen cyanide content (µg g*^−^*^1^) of cassava genotypes from different locations.

Variety	Cape-Coast	Ohawu	Fumesua	Mean ± SD
I090090	36.90	23.90	30.10	30.30 ± 6.50
I090151	32.20	26.00	36.30	31.50 ± 5.19
I070557	28.00	36.90	43.10	36.00 ± 7.59
I085392	26.00	47.80	9.90	27.90 ± 19.02
I083724	30.10	28.00	19.20	25.77 ± 5.78
I083774	23.90	37.40	26.00	29.10 ± 7.26
I070593	26.00	41.00	43.10	36.70 ± 9.33
I070539	28.00	41.00	-	34.50 ± 9.19
Cape Vars	47.80	41.00	30.10	39.63 ± 8.93
Local	28.00	41.00	30.10	33.03 ± 6.98
Mean ± SD	30.69 ± 7.04	36.40 ± 7.82	29.77 ± 10.72	

**Table 6 foods-09-01800-t006:** Total carotenoid content (µg g*^−^*^1^) of fresh and boiled cassava genotypes across three different locations, with percentage reduction after boiling (in parenthesis).

Fresh	Boiled
Variety	Cape-Coast	Fumesua	Ohawu	Cape Coast	Fumesua	Ohawu
I090090	6.11 ± 004	14.56 ± 0.04	10.00 ± 0.04	5.09 ± 0.04 (16.69)	11.29 ± 0.04 (22.46)	8.43 ± 0.04 (15.70)
I090151	5.98 ± 0.04	11.44 ± 0.04	8.52 ± 0.04	5.98 ± 0.04 (0.00)	10.64 ± 0.04 (6.99)	7.08 ± 0.04 (16.90)
I070557	11.99 ± 0.2	11.78 ± 0.07	9.87 ± 0.05	5.22 ± 0.09 (56.46)	11.49 ± 0.06 (2.46)	6.44 ± 0.04 (34.78)
I085392	4.80 ± 0.04	14.05 ± 0.04	11.51 ± 0.04	4.64 ± 0.04 (3.33)	10.51 ± 0.04 (25.20)	8.70 ± 0.04 (24.41)
I083724	4.63 ± 0.08	14.52 ± 0.04	11.70 ± 0.07	3.42 ± 0.04 (26.13)	13.86 ± 0.07 (4.55)	11.50 ± 0.04 (1.71)
I083774	6.81 ± 0.04	13.84 ± 0.04	9.89 ± 0.04	5.22 ± 0.04 (23.35)	11.81 ± 0.04 (14.67)	7.63 ± 0.04 (22.85)
I070593	8.15 ± 0.04	10.11 ± 0.04	18.81 ± 0.08	7.20 ± 0.04 (11.66)	9.21 ± 0.04 (8.90)	16.91 ± 0.06 (10.10)
I070539	7.81 ± 0.04	14.06 ± 0.04	15.74 ± 0.04	4.71 ± 0.2 (39.69)	12.38 ± 0.04 (11.96)	12.08 ± 0.04 (23.25)
Cape Vars	1.18 ± 0.04	1.34 ± 0.04	5.14 ± 0.08	1.01 ± 0.04 (14.41)	1.00 ± 0.04 (25.37)	4.74 ± 0.04 (7.78)
Local	1.49 ± 0.04	1.36 ± 0.04	4.90 ± 0.04	1.04 ± 0.04 (30.20)	1.02 ± 0.04 (25.00)	4.56 ± 0.08 (6.94)
Mean ± SD	5.87 ± 3.16	10.71+5.15	10.61 ± 4.27	4.27 ± 2.0 (27.26)	9.10 ± 8.85 (15.03)	8.85 ± 3.78 (16.59)
LSD (0.05) variety 2.52		*p* (variety)	<0.01	
LSD (0.05) treatment (fresh vs cooked) 1.13	*p* (treatment)	0.02

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
