# Peer review of "Proximate Composition, Cyanide Content, and Carotenoid Retention after Boiling of Provitamin A-Rich Cassava Grown in Ghana"

_foods, 2020, doi:10.3390/foods9121800_

Round 1

Reviewer 1 Report

This is the second revision of the manuscript entitled “Provitamin A - rich cassava proximate composition, cyanide content, and carotenoid retention after boiling”. I regret to say that my concerns have not been addressed sufficiently by the authors.

The results presented are not very innovative as a large number of studies have been conducted and published over time presenting similar results with different cassava genotypes. Therefore, I suggested that the study could have gained interest and relevance from a more rigorous analysis spanning over more than one single growth season. A deeper and critical analysis of the presented data, to find correlation among them (different environmental conditions of the different locations and different characteristics of the genotypes) and discussing them in relation to previously published results, would have added element of novelty and interest to the study.

I list below previous comments that do not have been addressed or commented in the second version.

  • Significant variations are observed for some parameters (not only HCN, and particularly crude fibres and ash content) among different growth locations, which in turns refer to different drought conditions. This should be analysed and compared in more details.
  • The authors should comment why despite it is reported that “cyanide content of cassava is higher during drought periods due to water stress in the plant [37]” this effect is not consistently observed in the dry cultivation area as compared to the humid location.
  • It is interesting to observe that the carotenoid content of some genotypes shows up to 2-3-fold difference in different locations. This has been partially highlighted but not commented.

Author Response

Please see attached the file in which we responded in a table format to the comments of all four the reviewers

Reviewer 2 Report

GENERAL IMPRESSION AND COMMENTS

This manuscript is about research is onto the effectivity of a cassava species which has an increased level of provitamine-A, to find out whether this is also available for the consumers of the products. The idea is to tackle a major problem in Ghana and many other African countries, malnutrition due to shortage of micronutrients in common foods. With cassava, a major issue is to process the raw food in such a way that the naturally present cyanide is reduced to a level that makes the processed food safe for human consumption and in the same time keep as much as possible of the carotenoids. This appears to be possible to a good extent, as it was concluded. The study is well designed and shows a good level of scientific soundness as the conclusions are well supported by the experimental data. The English is of good quality, texts are well readable. I have some questions and a few suggestions to improve the paper which can be consider 'minor revisions'.

A general question and comment is that for a non-food scientist the term 'biofortified' is perhaps to vague and needs a better explanation, both in the abstract and in other parts of the main text. Is there a natural variant of cassava that has yellow color due to higher carotenoids content which was further developed e.g. by selective breeding - or are these crops developed with genetic engineering methods and must be called GMO's? (Genetically Modified Organisms). Please state this clearly.

In this respect we must realize that there is still quite some  debate on the public acceptability of GMO-methods, perhaps more in Europe than elsewhere. Whether this issue should be discussed in a research paper is a question mark that I leave to the editor to decide on, based on the journal's policy.

DETAILED COMMENTS PER PARAGRAPH

The aim of this study is a bit hidden in the last lines of the introduction. I would like to see it just a little more pronounced, e.g. by making this a separate block within the introduction. Also, in this statement it would be good to write out abbreviated terms (TCC) in full.

In the tables, there are several entries with too many non-significant digits. For example, in Table 2 a value for carbohydrate content is reported as 26.7 ± 13.1 = this means, the error is about half the value of this parameter, but more important: at least the last digit of this example is completely non-significant and can be left out (e.g. 27 ± 13 would be more logic). Several more of such cases appear in this and other tables.

Since vitamin A and probably its precursors are fat-soluble, one might expect some correlation between fat content and TTC but it does not look like such in the experimental results. This question is not a condition for acceptance, it just triggers  scientific curiosity.

Reviewer 3 Report

The manuscript reports the proximal composition of nutrients in cassava roots; determinations include cyanide content, total protein, and total carotenoid levels of 8 biofortified cassava varieties. Cassava varieties were grown at three different Ghana locations, and authors reported total carotenoid retention during cassava boiling. 

Although data and literature reported are very relevant, the manuscript needs major revisions. 

Here some of my observations. 

  1. Cassava root sampling. It is not clear how many rows per plant and how many plants per row were harvested and bulked. Also, cassava roots were stored in a cool place, but temperature and the storage period's duration before the carotenoid determination is not reported. 
  2. The cooking condition of the cassava root is poorly described. More experimental details are needed; sample size, water temperature, and cassava cubes' dimensions were not reported. 
  3. Authors do not report using an internal standard during carotenoid extraction; therefore, it is an unknown % of carotenoid losses due to carotenoid extraction. 
  4. Although having a factorial experimental design (Treatments: Genotype and location), data analysis lacks  the use of appropriate statistical methods. Data was interpreted merely on the base of data ranges. 

Reviewer 4 Report

The paper entitled ‘Provitamin A - rich cassava proximate composition, cyanide content, and carotenoid retention after boiling’ deals with the compositional analyses of ten genotypes of yellow and white fleshed cassava obtained from different regions. This paper needs major revisions. Here are my detailed comments;

Abstract:

Lines 24-25: ‘However, the cyanide can be reduced in cassava genotypes with several processing methods such as roasting, frying, boiling, and fermentation’ Move this sentence to the introduction part.

Introduction:

Lines 58-59: Sentence ‘Good cooking … human consumption’ should be rephrased

Line 61: ‘water making up’?

Lines 61-64: These sentences should be re-arranged.

Line 70: ‘makes for’?

Lines 73-74: ‘in a way that can be measured’?

Lines 74-75: ‘Equally important is not to negatively affect agronomic characteristics of the crop, such as yield and disease resistance.’ Rephrase it.

Lines 120-122: These introductory sentence deal with ‘drought periods’. Based on this, authors should clearly point out the regions with ‘drought periods’ in the material and methods part.

Line 129: Add 2-3 lines about the clear objectives of this study.  

Materials and Methods:

Line 149: ‘small cubes’ mention size of the cubes

Line 153: ‘cooking in water’ how much was the solid to liquid ratio?

Lines 167-169: Briefly explain the protocol.

Line 174: ‘conical flask corked with filter paper by vacuum filtration’ not clear

Results:

Table 2

Lines 226-229: Make it clear which columns belong to ‘Moisture content’ & ‘Carbohydrate content’.

p-value <0.001, belongs to what?

Check all tables.

Table 4

‘±’ is missing in Ash content of I070557 from Cape Coast

Discussion:

Line 317-318 & 330-331: need revision

Line 338-340: Already mentioned in introduction.

Line 350-352: ‘Hence, micronutrient .. a positive impact’ delete it.

Line 357: ‘measured by spectrophotometer’ In the discussion part, there is no need to mention how it was measured.

Lines 358-359: ’was lower between ... across the three locations’ not clear

The relevant following publications could be mentioned or discussed in the manuscript;

Comparing Characteristics of Root, Flour and Starch of Biofortified Yellow-Flesh and White-Flesh Cassava Variants, and Sustainability Considerations: A Review. Sustainability 2018, 10, 3089; doi:10.3390/su10093089

Mc Dowell, I.; Oduro, K.A. Investigation of the ß-carotene Content of the Yellow Varieties of Cassava (Manihot esculenta Crantz). J. Plant. Foods 1983, 5, 169–171.

Vimala, B.; Thushara, R.; Nambisan, B.; Sreekumar, J. Effect of Processing on the Retention of Carotenoids in Yellow-Fleshed Cassava (Manihot esculenta Crantz) Roots. Int. J. Food Sci. Tech. 2011, 46, 166–169.

Ceballos, H.; Luna, J.; Escobar, A.F.; Ortiz, D.; Pérez, J.C.; Sánchez, T.; Pachón, H.; Dufour, D. Spatial Distribution of Dry Matter in Yellow Fleshed Cassava Roots and Its Influence on Carotenoid Retention upon Boiling. Food Res. Int. 2012, 45, 52–59.

Oliveira, R.G.A.; de Carvalho, M.J.L.; Nutti, R.M.; de Carvalho, L.V.J.; Fukuda, W.G. Assessment and Degradation Study of Total Carotenoid and  ß-carotene in Bitter Yellow Cassava (Manihot esculenta Crantz) varieties. Afr. J. Food Sci. 2010, 4, 148–155.

Thakkar, S.K.; Huo, T.; Maziya-Dixon, B.; Failla, M.L. Impact of Style of Processing on Retention and Bioaccessibility of ß-carotene in Cassava (Manihot escultenta Crantz). J. Agric. Food Chem. 2009, 57, 1344–1348.

Round 2

Reviewer 1 Report

The manuscript has improved and gained in clarity.

Newvertheless, I include two small suggestions

It is not clear whether one or two boiling procedures were used (lines 180-181 or 185-186). If one, the other description should be removed; if two, the reason for two different procedures should be given.

Lines 466-469, the sentence is a bit out of context. It should maybe be  rephrased to better clarify its sense.

Author Response

See uploaded file

Reviewer 4 Report

Authors should make it clear in the introduction or modify title in order to make it clear what is new in the present study as compare to the following published articles;

Ceballos, H.; Luna, J.; Escobar, A.F.; Ortiz, D.; Pérez, J.C.; Sánchez, T.; Pachón, H.; Dufour, D. Spatial Distribution of Dry Matter in Yellow Fleshed Cassava Roots and Its Influence on Carotenoid Retention upon Boiling. Food Res. Int. 2012, 45, 52–59.

Vimala, B.; Thushara, R.; Nambisan, B.; Sreekumar, J. Effect of Processing on the Retention of Carotenoids in Yellow‐ Fleshed Cassava (Manihot esculenta Crantz) Roots. Int. J. Food Sci. Tech. 2011, 46, 166–169.

Mc Dowell, I.; Oduro, K.A. Investigation of the ß‐carotene Content of the Yellow Varieties of Cassava (Manihot esculenta Crantz). J. Plant. Foods 1983, 5, 169–171.

Oliveira, R.G.A.; de Carvalho, M.J.L.; Nutti, R.M.; de Carvalho, L.V.J.; Fukuda, W.G. Assessment and Degradation Study of Total Carotenoid and ß‐carotene in Bitter Yellow Cassava (Manihot esculenta Crantz) varieties. Afr. J. Food Sci. 2010, 4, 148–155.

Thakkar, S.K.; Huo, T.; Maziya‐Dixon, B.; Failla, M.L. Impact of Style of Processing on Retention and Bioaccessibility of ß‐carotene in Cassava (Manihot escultenta Crantz). J. Agric. Food Chem. 2009, 57, 1344–1348.

Author Response

We have changed the title to:

Proximate composition, cyanide content, and carotenoid retention after boiling of provitamin A - rich cassava grown in Ghana

The results of this study are therefore specific to the growing conditions of Ghana.

This manuscript is a resubmission of an earlier submission. The following is a list of the peer review reports and author responses from that submission.

Round 1

Reviewer 1 Report

The work presented describe the evaluation of proximate characteristics in the roots of high-PVA cassava varieties grown in three separate locations in Ghana. It emphasizes on the cultural importance of processing cassava in sub-Saharan Africa and how these techniques can affect cyanide content (and therefore biosafety) but also pro-vitamin A retention with a main focus on boiling.

Overall the manuscript is sound and straight to the point, however, there is a large number of inconsistencies, inaccuracies and confusing statements throughout the text that will need to be addressed.

Please consider the below comments for improvement and refer to the .pdf document of my notes attached.

Thank you.

 Introduction

The introduction is long, convoluted and highly repetitive. I suggest spending some time to review it, restructuring it to make it more concise and straight to the point.

 Line 27-28

The following sentence should be modified: Total carotenoid content (TCC) of fresh and boiled cassava was 1.18 - 18.81 μg g-1 on a fresh weight basis, whereas in boiled cassava it was 1.01 - 13.36 μg g-1.”

 Also be careful of units used toward the text, concentrations should be displayed as µg.g-1 or µg/g but not ug g-1.

 Suggestion for change to: “Total carotenoid content (TCC) ranged from 1.18 - 18.81 μg.g-1 and 1.01 - 13.36 μg.g-1 (fresh weight basis) for fresh and boiled cassava, respectively.”

 Line 42

Reference 2 is too unspecific

Line 55

The reference to Sustainable Development Goal 2 comes out of the blue with no reference to this UN initiative. The authors might be familiar with this, but the readers might not. The inclusion of a reference or a link to the website would be very helpful.

Line 56

The term “hidden hunger” also come out of the blue and was not described previously. It should be.

Line 56-59

The sentence across those 3 lines needs referencing.

 The 2 paragraphs ranging from lines 60-82 need to be seriously looked through and re-written.

Some specific comments below:

 Lines 61-63

The following sentence is confusing “These genotypes have been tested across the various agro-ecologies in Ghana for their agronomic performance by the Crops Research Institute (CRI) towards possible commercial release in 2020.”

Suggestion for change to: “With the objective of releasing these genotypes in 2020, the Crops Research Institute (CRI) has been testing their agronomic performance across the various agro-ecologies in Ghana.”

Lines 68-71

This sentence is convoluted: “The roots consist mostly of starchy flesh (about 80% to 90% of the total weight of the root) with water making up a large proportion of this flesh [5]. The water content for cassava is in the range of 60.3% to 87.1% [6].”

What about: “The roots consist mostly of starchy flesh (80% to 90% by weight) with water making up 60.3% to 87.1% of its content [5, 6].”

Line 72

“reported to vary from 9.2% to 12.3% [7] and 11% to 16.5% [8].”

The data from these two references could be combined to show the published range only. The reader can get back to those references for more details.

Suggestion for change to: “reported to vary from 9.2% to 16.5% [7, 8].” or “reported to vary between 9.2% and 16.5% [7, 8].”

Line 75

Remove or in “of about or 1 - 3% on a dry mass basis”.

Line 76-77

Unit should be g.100g-1  or g/100g but not g 100-1 g

Part of the sentence is incorrect: “have about 10 g protein 100 g-1 fresh weight” should be “have about 10 g of protein per 100 g-1 fresh weight”.

Why is the starch content comparison made with maize and sorghum inferred from the levels of protein?

Wouldn’t it better to compare the levels of carbohydrates and starch directly?

Line 96

2 comments

1- pVAC is often the acronym for pro-vitamin A carotenoid in the literature, changing it here for provitamin A content could bring to confusions. It definitely was for me at first.

2- If it was decided to be kept as is, then the acronym should lose the “s”: as in pVAC and NOT pVACs.

Lines 113-116

Confusing. Difficult to understand the take home message from this sentence.

Lines 118

“Cyanogen” does this refer to hydrocyanic acid (HCN) only or to a group of cyanogenic compounds? If it is the later then the sentence should read: “the most important being cyanogens, which are responsible for the bitter taste of some cassava cultivars [24].”

Materials and Methods

Field trial design

Just would like to confirm that each location (3 in total) was laid in a randomized block design with 3 replicative blocks of 10 genotype each. Each genotype appeared in plots of 20 plants (laid as 4 rows of five plants). That is 200 plants per block and 600 per location. Total 1800 plants (all locations).

If this is correct, maybe a bit more details should be added to the section spanning lines 159 to 162 to make this clearer and emphasize the quantity of work that was done.

Sampling

1- My understanding of what is written is that for each block (or replicate) 1 sample per genotype was taken randomly. Was this sample taken from 1 single plant randomly selected from the 20 available? If this was the case I would argue that the plot level variation was lost in the process and at least a composite sample made of roots from 3 to 5 plants (out of 20) from each plot would have been a better option.

2- Line 172 refer to matching sets of samples that were boiled. Although a simple process, please include a sentence describing how that was done here rather than in lines 208-209?

 Results

3.1. Moisture content

Line 219

Range for Fumesua is 54.8% (Cape Vars) to 80.07.

3.2. Carbohydrate content

Line 229-230

“The local (white-fleshed) and improved variety (Cape Vars) recorded higher carbohydrate content than most of the yellow-fleshed cassava genotypes.”

 Confusing, does this sentence refer to Cape Vars only or to Cape Vars and Husivi? If it is Cape Vars only than the sentence is redundant to the previous sentence and unnecessary. Anyhow, the statement should be “recorded higher carbohydrate content than ALL of the…”

 3.3. Protein and fat contents and 3.4. Crude fibre and ash content

Line 243

Fat proportion for I070557 at Ohawu is 1.14% not 1.16% according to Table 3.

Table 3 and 4

Titles refer to the data presented as being “protein, fat …content”, it is not. The data presented represent percentages of protein, fat…content.

Table 4

The “Crude fiber” heading is missing the unit (%).

3.5. Hydrogen cyanide content

 All references to genotypes in the text have now changed from the prefix “I” to “IBA”. Example I083774 is now IBA083774.

Line 269

The statement “The HCN of samples differed significantly (p>0.05).” Is inaccurate on two counts. First for samples to be statistically different, their p value has to be inferior to 0.05 (p<0.05, for 95% confidence). Second Table 5 shows two confusing p values, none of which are inferior to 0.05. I assume that p= 0.2 refer to a mean comparison between the HCN values of each locations. This indicates that there are no statistical differences between locations.

Line 271

There is a reference to a Figure 2. I assume it should be Table 5 instead.

Line 279

The value for Fumesua from Table 5 is 29.77 mg.kg-1 and not 29.73 mg.kg-1.

Lines 280-281

“(30.00±7.04, 37.78±7.02, 29.73±10.02 at P=0.10)” these values have no units and do not appear in Table 5 either.

3.6. Total carotenoid content

 This section has a number of issues.

1- Starting with the units for TCC. The convention in most publications is µg.g-1.

Table 6 uses mg.kg-1, which is the same, however the text uses µg.g-1. Please choose one and be consistent through the document.

2- The text never refers to Table 6.

Line 291

18.81 µg.g-1 is for sample I070593 not I070539.

Line 293

The boiled samples top TCC range is 16.91 µg.g-1, also for sample I070593 and not 13.86 µg.g-1.

Line 297

Two data points are missing their units (µg.g-1).

Convention is “p” that refers to p values to be italicized as in p= 0.02.

Table 6

A few comments:

- p-values displayed are the results of the ANOVA across genotypes. However, I cannot see the result of the Post Hoc test showing statistical differences between lines.

- Since the importance of this work is to also understand the potential losses of TCC through processing methods, I would have liked to see the difference in TCC between fresh and boiled expressed as a percentage of reduction or increase. This would make it easier to see any trends. In addition, is the difference seen between fresh and boiled significant? This was not determined.

Discussion

 - The first paragraph is full of values not in accordance with the respective Tables form the Result section (see .pdf notes).

- Line 343-347

Some interesting points. Boiling can indeed have a positive impact on carotenoid retention. If it is not done for too long, boiling could help release carotenoids from the complex matrix of starchy crops. This results in carotenoids more easily extracted and available for measurements but also more available for nutrition. In addition, gentle cooking has an established benefit on bioavailability of carotenoids and bioconversion into retinol.

- A very important point that was only briefly touch on is whether or not any of these improved “high” carotenoid varieties have TCC levels high enough to be nutritionally significant? On average (and in Ghana for example) how much percentage of the EAR for vitamin A each of these varieties could contribute? And how are these numbers affected by boiling? These are very simple calculations that can be done with minimal assumptions (consumptions, bioavailability). These would definitely bring more depth to this work and the usefulness of these varieties.

- All of these new “high TCC” varieties have reduced DMC. This is expected in PVA-biofortified cassava varieties whether they have been generated through conventional or molecular breeding techniques. There are actually underlying physiological reasons for this. Although this was mentioned a little bit in the conclusion, I would have like to see some discussion around DMC and the potential impact on processing and therefore adoption of these varieties.

Reviewer 2 Report

In the manuscript entitled “Provitamin A - rich cassava proximate composition, cyanide content, and carotenoid retention after boiling” the authors compare composition (including carotenoid content of fresh and boiled roots) and cyanide content of eight yellow-fleshed cassava genotypes and two (one commercial and one Ghanaian landrace) white fleshed variety, cultivated in three agroecological zones in Ghana, for one season.

The paper is clearly written, albeit the introduction should be more concise as several statements are repetitive. Long and redundant literature citations (e.g. lines 318-329) are also included in the discussion, which instead would benefit from a more detailed comparison of the reported results with previously published data.

Unfortunately, the results presented are not very innovative as a large number of studies have been conducted and published over time presenting similar results with different cassava genotypes. However, a deeper analysis of the presented data and correlation analyses among them would certainly add element of novelty and interest to the study.

As follow I highlight some points that could be addressed for manuscript improvement.

  • More reliable conclusion could be drawn after repetition of the experiments for more than one single growth season.
  • Root samples were collected from apical, middle and distal portions of the roots. It would be interesting to find out whether different values can be correlated to specific root portions
  • Significant variations are observed for some parameters (not only HCN, and particularly crude fibres and ash content) among different growth locations, which in turns refer to different drought conditions. This should be analysed and compared in more details.
  • The authors should comment why despite it is reported that “cyanide content of cassava is higher during drought periods due to water stress in the plant [37]” this effect is not consistently observed in the dry cultivation area as compared to the humid location.
  • In Table 2 and in the corresponding text (lines 225 and 226) it should be stated that this is “calculated carbohydrate content”.
  • It is interesting to observe that the carotenoid content of some genotypes shows up to 2-3-fold difference in different locations. This should be highlighted and commented.